# Physiological and Biochemical Traits in Korean Pine Somatic Embryogenesis

**Chunxue Peng [1], Fang Gao [1], Hao Wang [1], Hailong Shen [1,2,\*] and Ling Yang [1,2,\*]**

[1] State Key Laboratory of Tree Genetics and Breeding, School of Forestry, Northeast Forestry University, Harbin 150040, China; chunxue@nefu.edu.cn (C.P.); fanggao@nefu.edu.cn (F.G.); 15636830205@163.com (H.W.)

[2] State Forestry and Grassland Administration Engineering Technology Research Center of Korean Pine, Harbin 150040, China

\* Correspondence: shenhl-cf@nefu.edu.cn (H.S.); yangl-cf@nefu.edu.cn (L.Y.)

**Abstract:** Korean pine broadleaf mixed forest is an important ecosystem for maintaining biodiversity in Northeast China. Korean pine is also an important species for the production of timber and nuts in the mountainous areas of Northeast China. In this study, we compared three types of Korean pine callus and found that embryogenic callus had high amounts of storage substances (protein, sugar and starch). Non-embryonic callus had high levels of polyphenols and polyphenol oxidation, while callus that lost somatic embryogenesis potential had lower levels of storage substances (protein, sugar and starch) and higher contents of peroxidase and catalase. These results indicate that high contents of storage substances (protein, sugar and starch), and low levels of polyphenols and polyphenol oxidase can be used as physiological markers of callus with somatic embryogenic potential. During the development process of Korean pine somatic embryos, fresh weight and dry weight gradually increased, while water content gradually decreased. Soluble protein, starch, soluble sugar and superoxide dismutase also increased during development, while peroxidase and catalase levels reduced over time. These results indicate that somatic embryogenesis involves energy storage, and antioxidant enzymes cooperate to regulate the occurrence and development of embryos. These results provide physiological markers for identification of embryogenic callus with somatic embryogenesis, to evaluate callus suitable for somatic embryogenesis, and provide basis for further research on the molecular mechanisms of somatic embryogenesis.

**Keywords:** *Pinus koraiensis* Sieb. et Zucc.; somatic embryogenesis; non-embryogenic callus; embryogenic callus; resistance oxidase activity

## 1. Introduction

Conifers are distributed worldwide, dominating large forest ecosystems and playing essential roles in global carbon fixation as well as biodiversity maintenance. Conifers are also economically important since they are utilized for a vast range of products, including wood, pulp, biomass and diverse secondary metabolites [1]. Korean pine (*Pinus koraiensis* Sieb. et Zucc.) is the dominant species of Korean pine forests, where it plays an important ecological role. Such forests are widely distributed in Northeast China, southeastern Russia, South Korea, North Korea and Japan [2]. Korean pine has been listed as an endangered species in China, primarily due to overharvesting. The breeding history of Korean pine is short, and the number of seeds in seed repositories is limited.

As an important means of plant regeneration, somatic embryogenesis technology has the potential to enable low-cost plant expansion and the rapid production of somatic embryo (SE) seedlings with high genetic stability [3]. This process not only provides a model for studying different developmental,

molecular and biochemical processes [4] but could also provide suitable target material for genetic transformation that enables the generation and propagation of plants with desirable genetics [5]. Therefore, understanding the biological mechanism of somatic embryogenesis lays the foundation for further research on the molecular mechanisms of somatic embryogenesis. The callus of Korean pine has been successfully obtained from mature zygotic embryos as well as immature zygotic embryos [6,7], with successful plant regeneration from immature zygotic embryos [8]. However, there are still some problems that limit the practical application of the Korean pine SE. In earlier studies, it was found that the callus obtained from the mature zygotic embryo could not produce SEs in many cases [7]. Tissue composition may be responsible for the failure of further development of the embryogenic lines, as the mature seed-derived callus cells contain high levels of lethal phenolics and oxidative compounds that possibly arrest further development of the callus cells. Additionally, too high levels of 2.4-dichlorophenoxyacetic acid (2.4-D) may cause cells to mimic embryogenic cell behavior without truly converting them [9,10]. The embryogenic callus (EC) obtained from the immature zygotic embryo could produce SEs [11], but the SE production rate was low and root growth was limited. In addition, EC may lose the potential for somatic embryogenesis after extended subculture. Issues surrounding this process hinder large-scale propagation of somatic embryogenesis techniques. Therefore, molecular or biochemical markers are needed to evaluate callus suitable for somatic embryogenesis as a first step towards solving the problems associated with this process in pine trees.

Silveira et al. [9] found that embryogenic callus is smooth and dense in sugarcane, while non-embryonic callus is brittle or soft and translucent. Nieves et al. [10] and Mahmud et al. [12] compared metabolites between sugarcane embryogenic and non-embryonic callus tissues including fats, soluble proteins, amino acids, major carbohydrates, organic acids, phenols and polyamines. According to their results, many of these metabolites have an impact on the somatic embryogenesis process. In a study by Mangosteen, it was found that the embryonic globular structure and the embryogenic nodular compact structure can be used as morphological markers of embryogenesis. A large number of amino acids, as well as mannose, can act as metabolic biomarkers in embryogenic tissues [13]. Therefore, identifying the morphological and physiological differences between embryogenic callus and non-embryonic callus is essential for improving the design of the somatic embryogenesis process.

Somatic embryogenesis is a complex developmental process, which includes cell division, differentiation and a series of physiological changes [14,15]. During somatic embryogenesis of *Silybum marianum* L., mature somatic embryos were found to contain the highest levels of carbohydrates, starch, ascorbic acid and total free amino acids [16]. In the process of Persian walnut somatic embryogenesis, it was found that Catalase (CAT), Peroxidase (POD), Polyphenol oxidase (PPO) and Superoxide Dismutase (SOD) can not only be used as quantitative and qualitative indicators of the developmental stage of somatic embryos, but also reflect the difference between in vivo and in vitro conditions. At the same time, it was also found that translucent cotyledon embryos lack starch granules in epidermal cells compared to opaque cotyledons, while torpedo-shaped somatic embryos contain the highest levels of proline, protein and antioxidant enzyme activity [15]. This demonstrates that understanding the dynamic changes in key biomarkers during the process of somatic embryo differentiation can also reveal the biochemical regulatory mechanisms taking place. Despite the importance of understanding this process, no report has been published on somatic embryogenesis in Korean pine. Soluble protein, soluble sugar and starch content are key indicators of physiological and biochemical changes during plant growth and development [17]. As catalysts for plant physiological and biochemical reactions, enzymes play important roles in somatic embryogenesis. In particular, enzymes related to reactive oxygen metabolism (SOD; POD; CAT; etc.,) play crucial roles in cell division and differentiation [18,19]. Studying the morphological and physiological changes during somatic embryogenesis is important for understanding the molecular response mechanisms that influence SE development and maturation.

Here, we observed morphological differences in Korean pine EC, non-embryonic callus (NEC), and callus which had lost somatic embryogenesis potential (E-L). The differences in soluble protein, soluble sugar, starch, proline, polyphenol, PPO and antioxidant enzymes (SOD, POD and CAT) of

the three callus types were determined. The goal of this study is to provide morphological and physiological markers which are correlated with the somatic embryogenesis potential of callus. In addition, morphological and physiological changes which occur in embryogenic cultures during the maturation of Korean pine SEs are examined. The fresh weight (FW), dry weight (DW), water content, soluble protein, soluble sugar, starch, antioxidant and enzyme activities (SOD, POD and CAT) of cultures at different developmental stages were also determined. The biomarkers verified in this study lay a foundation for understanding the molecular mechanisms of somatic embryogenesis in Korean pine.

## 2. Materials and Methods

### 2.1. Source of Plant Material and Embryogenic Culture

Immature zygotic embryos were used to initiate EC in Korean pine. DCR medium [20,21] for EC induction medium, which contained 2.0 mg $L^{-1}$ 1-Naphthaleneacetic acid (NAA), 1.5 mg $L^{-1}$ 6-benzyl amino-purine (6-BA), 0.5 g $L^{-1}$ casein hydrolysate (CH), 0.5 g $L^{-1}$ L-glutamine, 30 g $L^{-1}$ sucrose and 4 g $L^{-1}$ gellan gum (Phytagel™, Sigma-Aldrich, St. Louis, Mo., USA). After about one month of exposure to explant induction culture, EC was extruded from the bead end of the female gametophyte (Figure 1a). The cell line ECM28-10 was used in the study. The NEC was derived from mature zygotic embryos, using the same medium used for the initiation of EC. About half a month after explant culture, NEC grew on the cotyledon edge of the zygotic embryo (Figure 1f). The cell line NECM28-10 was used in the study. In this study, immature seeds and mature seeds were obtained from the same open-pollinated mother tree. Embryogenic callus has been shown to lose some somatic embryogenesis potential after 9 months of subculture [11]. The cell lines marked as E-LM28-10 represented original EC that lost maturation capacity after prolonged in vitro culture on DCR medium (Figure 1h). To promote callus proliferation, EC, NEC, and E-L were transferred to proliferation medium (Figure 1b,j,h), which consisted of DCR supplemented with 0.5 mg $L^{-1}$ 2,4-D, 0.1 mg $L^{-1}$ 6-BA, 30 g $L^{-1}$ sucrose, 0.5 g $L^{-1}$ L-glutamine and 0.5 g $L^{-1}$ CH. The pH of the medium was then adjusted to 5.8 before sterilization. EC, NEC, and E-L were subcultured at intervals of 2 weeks to maintain and proliferate, with the culture placed in darkness at 25 °C. The EC, NEC, and E-L after two passages were used in this study.

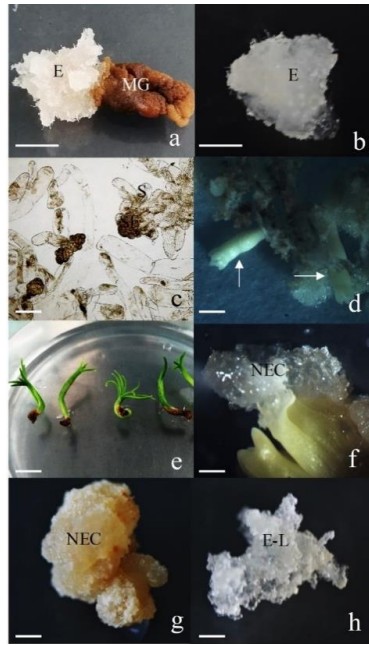

**Figure 1.** Characterization of Korean pine embryogenic callus, non-embryogenic callus and embryogenic

callus that lost the maturation capacity. (**a**) Immature seeds start embryogenic callus after being cultured for 1 month in the induction culture period, E—the embryogenic callus was protruded from micropylar end of the megagametophyte (MG) explant, bar = 0.5 cm; (**b**) Embryogenic callus after two subcultures, bar = 0.5 cm; (**c**) Microscopic observation of embryogenic callus, SE—early somatic embryo, S—organized into bundle, bar = 100 μm; (**d**) Cotyledonary somatic embryos, bar = 0.5 cm; (**e**) Regenerated emblings, bar = 0.5 cm; (**f**) Initiation of non-embryogenic callus, bar = 0.1 cm; (**g**) Non-embryogenic callus after two subcultures, bar = 0.5 cm; (**h**) Callus that has lost embryogenesis ability after 9 months of subculture, bar = 0.5 cm.

## 2.2. Maturation of Somatic Embryos

The ECM28-10 cell line, which was subcultured on the proliferation medium 2 times, was used for further experiments. To stimulate the development of early SE, EC was harvested after 1 week of subculture to induce somatic cell maturation. This maturation medium was based on mLV [22] supplemented with 80 μM abscisic acid (ABA), 68 g L$^{-1}$ sucrose, 0.5 g L$^{-1}$ casein hydrolysate, 0.5 g L$^{-1}$ L-glutamine and 12 g L$^{-1}$ gellan gum. The culture was grown in darkness at 23 ± 1 °C for 10–12 weeks.

## 2.3. Microscopic Observation

We used a microscope to observe the morphology of embryogenic callus obtained from immature seeds, non-embryonic callus obtained from mature zygote embryos and callus which had lost somatic embryogenesis potential. EC was collected on the 7th day of proliferation culture, stained with 2% acetyl-carmine, compressed on slides, covered with cover glass, and observed under a Zeiss A× microscope (Carl Zeiss, Jena, Germany). In addition, we also observed the morphological changes during somatic embryogenesis of Korean pine under a microscope (SZX-ILLB2-200, Olympus Corporation, Tokyo, Japan). The developmental period of SEs of Korean pine was defined according to the results of Von Arnold et al. [23]. Materials were collected for morphological observation at 1, 2, 3, 4, 5 and 8 weeks of somatic embryo maturation and 2 months of somatic embryo germination.

## 2.4. Sample Collection

For EC, NEC, and E-L, three biological replicates were collected on the 7th day of subculture, and 0.5 g of callus was collected for each sample for physiological examination. ECs were collected for somatic embryo maturation on the 7th day of subculture. During somatic embryo maturation, three biological replicates were collected at 0, 1, 2, 3, 4 and 5 weeks, with 0.5 g obtained for each sample for physiological analysis. These samples were then wrapped in aluminum foil, immediately frozen in liquid nitrogen and stored at −80 °C until further analysis.

## 2.5. FW, DW and Water Content of EMs

Fresh weight, dry weight and water content were measured at 0, 1, 2, 3 and 4 weeks of maturation. Fresh weight (FW) was obtained by measuring the weight of filter paper with dispersed cultures, then subtracting the mass of the filter paper. Dry weight (DW) was determined after drying in an oven at 70 °C for 24 h. The moisture content was calculated as (FW-DW) / DW and expressed in g $H_2Og^{-1}$ DW [24]. Each measurement was replicated 10 times.

## 2.6. Physiological Determinations

### 2.6.1. Soluble Sugar Content

Extraction of total sugar was carried out according to the method of Somani and Businge [25], with minor modifications. Approximately 0.5 g of fresh sample was ground in a mortar with liquid nitrogen. The sample was boiled in 5 mL 80% ethanol at 80 °C in a heating block for 30 min, then centrifuged at 12,000 g for 15 min, and 1 mL of supernatant was taken. Soluble sugar content was

determined with the sulfuric acid anthrone method and measured by a spectrophotometer (Ultrospec 2100 pro, *Amersham Pharmacia Biotech*, Little Chalfont, UK) at 620 nm measured.

### 2.6.2. Starch Content

Extraction of starch was modified according to the method of Vale et al. [26]. The particles from soluble carbohydrate extract were resuspended in 1 mL of perchloric acid (9.2 mol·L$^{-1}$), stirred for 15 min, and then centrifuged at 12,000 g for 10 min at 4 °C and the supernatant was collected. The resulting precipitate was then resuspended in 1 mL of perchloric acid (4.6 mol·L$^{-1}$), stirred for 15 min, and centrifuged at 12,000 g for 10 min at 4 °C, and the supernatant was collected. The two collected supernatants were mixed and brought up to 50 mL. Starch quantification proceeded according to the method described by Colvin et al. [27] with minor modifications, the samples (1 mL) were mixed with 5 mL of 0.1% anthrone. This mixture was boiled in boiling water at 100 °C for 10 min. After cooling, the spectrophotometer reading was taken at 620 nm. Starch concentrations were determined using a range of glucose concentrations as standards and multiplying the readings by 0.9 according to McCready et al. [28].

### 2.6.3. Soluble Protein Content

Soluble protein was determined by the method described by Liu et al. [29], with minor modifications. Approximately 0.5 g of fresh sample was homogenized in 5 mL phosphate buffer solution (pH 7.0). The homogenate was centrifuged at 12,000 g for 15 min at 4 °C, and 0.1 mL of the supernatant was added to 5 mL of Coomassie brilliant blue G-250 solution (0.1 g L$^{-1}$). After 2 min, the soluble protein content was determined by measuring absorption at a wavelength of 595 nm.

### 2.6.4. Polyphenol Content

Polyphenol content was determined according to the method of Kaewubon et al. [30], with minor modifications. Approximately 0.5 g of fresh samples were homogenized in 5 mL of 0.1 M sodium phosphate buffer (pH 7.2). The homogenate was centrifuged at 12,000 g for 15 min at 4 °C. A total of 50 μL of the supernatant was added to 250 μL of Folin-Ciocalteu phenol reagent and 750 μL 10% $Na_2CO_3$. After 15 min of incubation at 25 °C in the dark, the absorbance at 750 nm was measured.

### 2.6.5. Proline Content

Proline content was determined according to the ninhydrin method of Helaly et al. [31] with minor modifications. Approximately 0.5 g of fresh sample was homogenized in 5 mL of 3% sulfosalicylic acid solution. The homogenate was centrifuged at 12,000 g for 15 min at 4 °C. A total of 0.1 mL of the supernatant was added to 2 mL of acid ninhydrin and 2 mL of glacial acetic acid, followed by boiling for 60 min. The mixture was then treated with toluene, and free proline was quantified spectrophotometrically by measuring the absorption at 520 nm.

### 2.6.6. Antioxidant Enzyme Activities

For the assessment of antioxidant enzymes, approximately 0.5 g of fresh samples were homogenized in 5 mL of 0.1 M sodium phosphate buffer (pH = 7.8). The homogenate was centrifuged at 12,000 g for 15 min at 4 °C. Superoxide dismutase (SOD) activity was estimated by measuring its ability to inhibit the photochemical reduction of nitroblue tetrazolium (NBT) at 560 nm as described by Jariteh et al. [15]. A total of 50 μL of the supernatant was added to 3 mL of reaction mixture (0.1 mM EDTA, 50 mM sodium phosphate buffer (pH 7.8), 75 μM NBT, 13 mM L-methionine and 75 μM riboflavin). After 15 min of incubation at 25 °C in the dark, the absorbance at 750 nm was measured. Peroxidase (POD) activity was measured according to the method of Rahnama et al. [32]. A total of 0.2 mL of the supernatant was added to the reaction mixture (4 mL of 0.2 M acetate buffer (pH = 4.8), 0.4 mL of 3% $H_2O_2$ and 0.2 mL of 20 mM benzidine). The increase in absorbance was

then recorded at 530 nm and the POD activity was defined as 1 μM of benzidine oxidized per min per mg protein (unit mg protein). Catalase (CAT) activity was measured according to the method of Jariteh et al. [22]. A total of 50 μL of the supernatant was added to 0.7 mL reaction mixture (0.625 mL of 50 mM sodium phosphate buffer (pH = 7) and 0.075 mL of 3% $H_2O_2$). The activity of CAT was determined from the $H_2O_2$ decomposition rate, which was measured by a decrease in absorbance at 240 nm. Polyphenol oxidase (PPO) activity was measured according to the method of Kaewubon et al. [30], with minor modifications. Approximately 0.5 g of fresh sample was homogenized in 5 mL of 0.1 M sodium phosphate buffer (pH = 7.2). The homogenate was centrifuged at 12,000 g for 15 min at 4 °C. A total of 20 μL of the supernatant was added to 2.7 mL of the reaction mixture (2.5 mL of 0.2 M sodium phosphate buffer (pH = 6.8), 0.2 mL of 20 mM pyrogallol). The increase in absorbance was recorded at 430 nm and the PPO activity was defined as 1 μM of pyrogallol oxidized per min per mg protein (unit mg protein).

### 2.6.7. Statistical Analysis

All data were analyzed by one-way analysis of variance (ANOVA) and Duncan's multiple comparisons using SPSS software (2010, V. 19.0: SPSS, Inc., Cary, NC, USA), and plotted with Sigmaplot (v12.5, SYSTAT, San Jose, Calif., USA) software and Microsoft Visio 2007 software.

## 3. Results

### 3.1. Characterization of Analyzed Tissues

We observed that the EC was white, transparent, soft and sticky, with a filamentous structure on the surface (Figure 1b). Microscopic observation revealed early SEs were present (Figure 1c). The transfer of EC to the maturation medium was sufficient to induce cotyledonary somatic embryos (Figure 1d). NECs were initiated on mature zygotic embryos (Figure 1f). The NECs were white, translucent and fragile, with a granular structure. After several subcultures, the texture became hard and the color changed to light yellow (Figure 1g). The embryogenic cell line EM28-10, which had been cultured for one year, lost maturation ability and was designated as E-L. The color of E-L was white and translucent, with a soft fragile appearance (Figure 1h).

### 3.2. Physiological Differences in Different Types of Callus

There were significant differences in soluble protein, soluble sugar, starch, proline, polyphenols, PPO, POD, SOD and CAT among Korean pine EC, NEC and E-L ($p < 0.05$). Compared with NEC, EC had higher soluble protein, soluble sugar, starch and proline content, and lower polyphenol, POD, SOD and POD activities. Compared with E-L, EC had higher soluble protein, soluble sugar and proline content, and lower POD and CAT activity. However, there was no significant difference between starch content, polyphenol content, PPO and SOD activity. Compared with E-L, NEC had higher polyphenol content, PPD and SOD activity, lower starch content and CAT activity, but no significant difference in soluble protein, soluble sugar, proline content or POD activity (Figure 2).

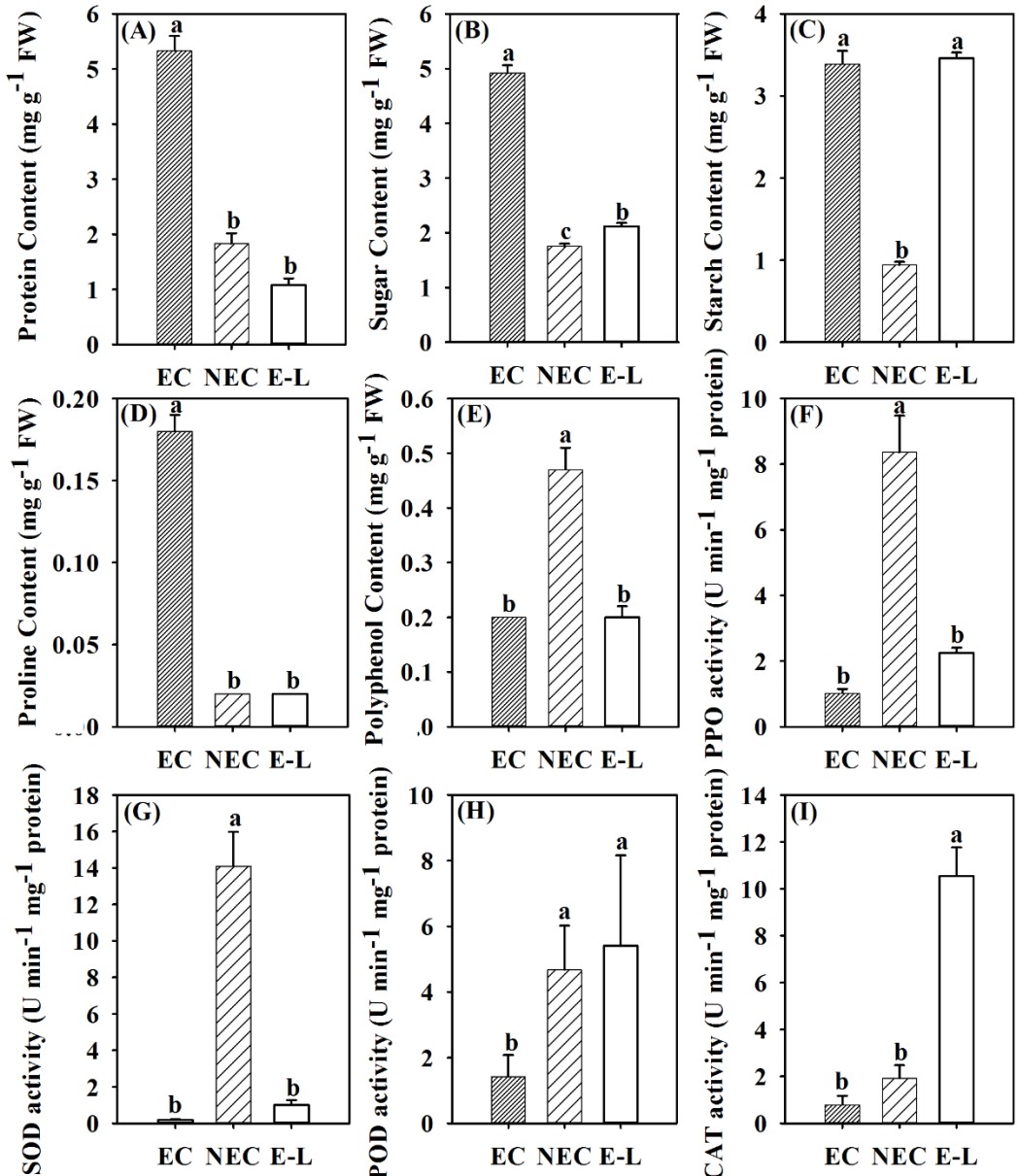

**Figure 2.** Physiological differences between different callus tissues of Korean pine on the 7th day of culture on proliferation medium. (**A**) Protein content, (**B**) Sugar content, (**C**) Starch content, (**D**) Proline content, (**E**) Polyphenol content, (**F**) PPO activity, (**G**) SOD activity, (**H**) POD activity, (**I**) CAT activity. EC: Embryogenic callus; NEC: Non-embryogenic callus; E-L: callus which has lost somatic embryogenesis potential. Different letters above bars indicate significant differences between samples. (Duncan's test; $P = 0.05$).

*3.3. Morphological Changes during Somatic Embryogenesis in Korean Pine*

After 1 week of culture on the mLV maturation medium, immature embryos with white transparent filaments were observed (Figure 3a1). After 8 weeks of culture, these immature embryos had expanded and developed into fully mature embryos. Embryos passed through the early embryo stage (1–3 weeks), the late embryo stage (3–4 weeks), the early cotyledon stage (5–6 weeks) and the late cotyledon stage (8 weeks) (Figure 3a2–a6).

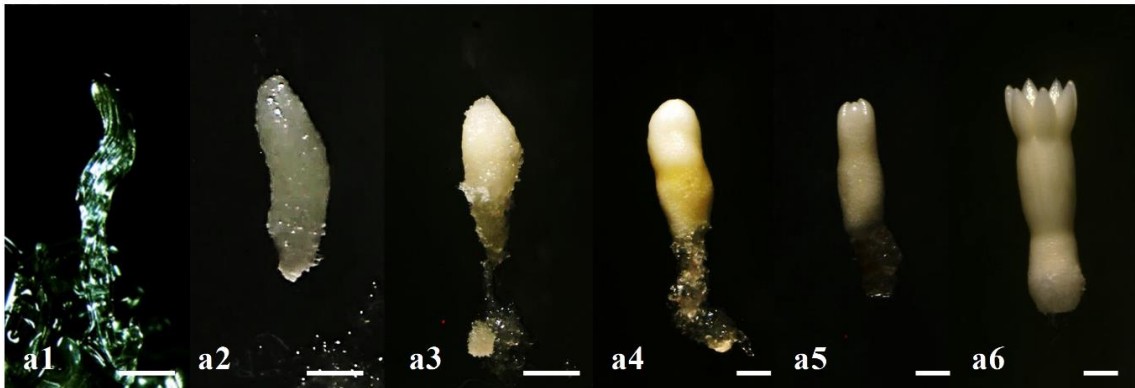

**Figure 3.** Korean pine somatic embryo development. (**a1**) Somatic embryos cultured on maturation medium for 1 week. (**a2**) Somatic embryos cultured on maturation medium for 2 weeks. (**a3**) Somatic embryos cultured on maturation medium for 3 weeks. (**a4**) Somatic embryos cultured on maturation medium for 4 weeks. (**a5**) Somatic embryos cultured on maturation medium for 5 weeks. (**a6**) Somatic embryos cultured on maturation medium for 8 weeks (Bar = 1 mm).

*3.4. Fresh Weight, Dry Weight and Water Content of EC*

During the SE maturation of Korean pine, the FW, DW and water content of EC were significantly different ($p < 0.05$). FW and DW gradually increased with culture time, while the water content decreased. Compared with week 0, ECs at 5 weeks of maturity had higher FW (6.8 fold), DW (29 fold) and lower water content (9.39 g $H_2O$ g$^{-1}$) (Figure 4).

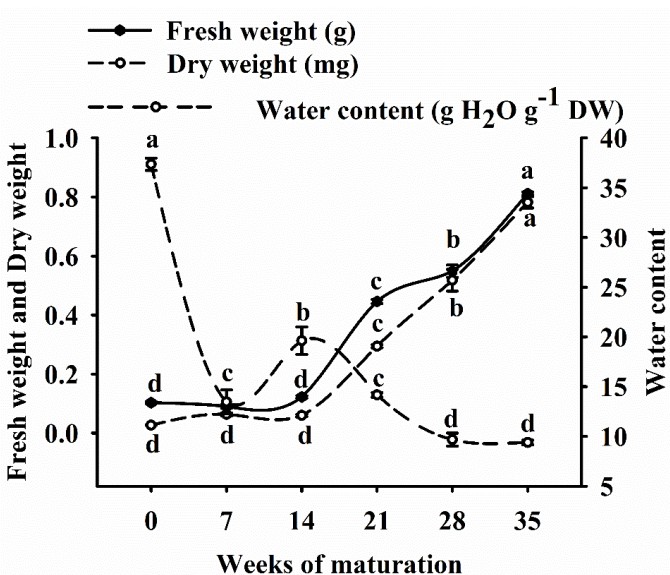

**Figure 4.** Changes in fresh weight, dry weight and water content of the Korean pine somatic embryos during maturation. Different letters above the curve indicate significant differences between samples. (Duncan test; $P = 0.05$).

*3.5. Levels of Soluble Sugar, Soluble Protein and Starch*

The highest level of soluble sugar (10.40 mg g$^{-1}$ FW) was detected in the second week of embryogenesis, which was 1.78 times higher than that in week 0. There was no significant change in soluble sugar content from week 2 to week 5. The highest levels of soluble protein (10.72 mg g$^{-1}$ FW) and starch (12.52 mg g$^{-1}$ FW) were detected in the 5th week of embryogenesis. Compared to week 0, the soluble protein and starch content increased by 3.95 and 3.82 fold (Figure 5).

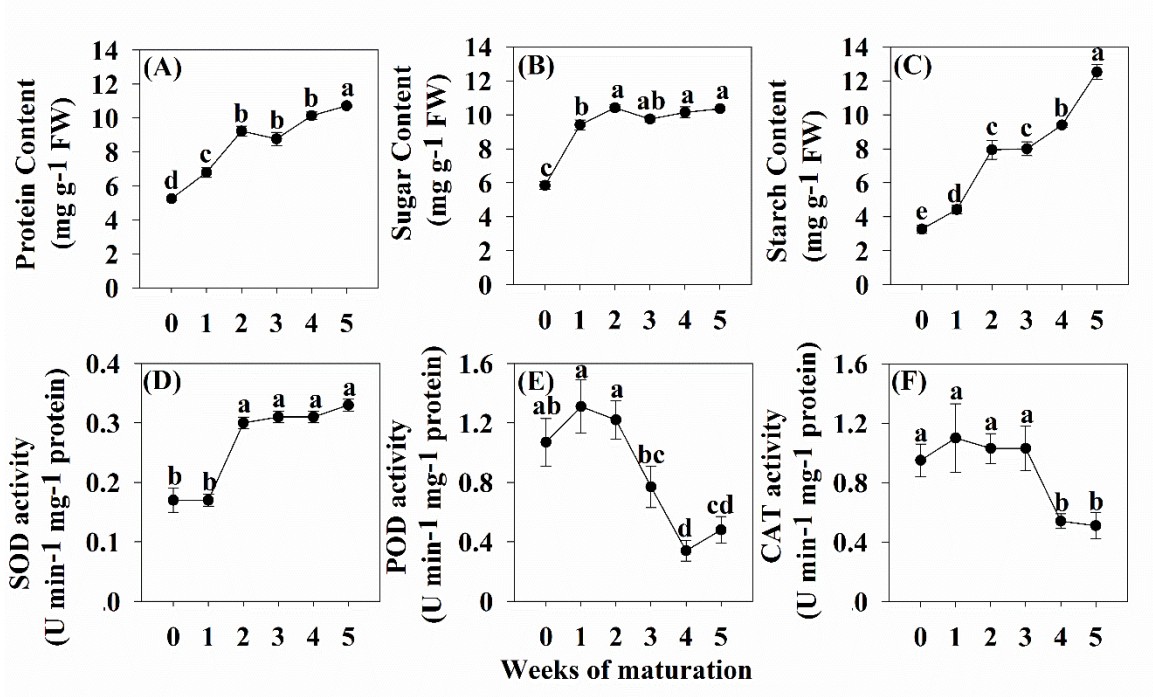

**Figure 5.** Physiological changes of somatic embryos of Korean pine in maturation medium at 0, 1, 2, 3, 4 and 5 weeks. (**A**) Sugar content, (**B**) Starch content, (**C**) Proline content, (**D**) SOD activity, (**E**) POD activity, (**F**) CAT activity. Different letters above the curve indicate significant differences between samples. (Duncan test; $P = 0.05$).

### 3.6. Levels of SOD, POD and CAT Activity

During the maturation of SEs, SOD activity increased significantly at the early stage of somatic embryo development (week 1 to week 2). At its peak, SOD activity was as high as 0.33 U min$^{-1}$ mg$^{-1}$ protein, an increase of 78.41% over week 0. The SOD activity was consistently high from early somatic embryos to cotyledonary embryos (week 2 to week 5). POD activity was high during the early somatic stage (week 0 to week 2), and gradually decreased during the transition from early somatic stage to cotyledonary embryo (week 2 to week 5). POD activity at week 4 was reduced by 68.22% compared to week 2 (0.34U min$^{-1}$ mg$^{-1}$ protein). CAT activity was high during early somatic development (week 0 to week 3), began to decline at week 3, and dropped to 0.54 U min$^{-1}$ mg$^{-1}$ protein by week 4 (Figure 5).

## 4. Discussion

### 4.1. Physiological Differences between Different Types of Callus

Both the initiation and the maintenance of EC are important for somatic embryogenesis [9]. In the induction of EC, the source of explants is considered to be the most important factor, as the response of EC is strongly dependent on age, location, time and genetic characteristics [33]. In our study, we observed that callus obtained from mature seeds had no potential for somatic embryogenesis, while EC obtained from immature seeds lost SE production ability after one year of subculture. This effect was also observed in *Pinus nigra* and peach palm (*Bactris gasipaes* Kunth) [34,35]. Studies have pointed out that the physiological factors of callus are very important for somatic embryogenesis [9,36]. To obtain insight into the processes that may lead to loss of embryogenic ability, we compared the physiological differences between EC, NEC and E-L. Starch and sugar can be used as a source of energy and are critical to development [37]. In our study, EC had higher levels of soluble protein and soluble sugar compared with NEC and E-L. Similar results have been reported in *Catharanthus roseus* (Linn.) G. Don and sugarcane [10,38]. High carbohydrate concentrations appear to play a key role in embryogenesis

and embryo development [12]. According to some reports, EC has strong metabolic activity and can provide energy for the further development of SEs [39]. In our study, Korean pine EC has high metabolic activity and can further differentiate into cotyledon-type SEs, while NEC and E-L have lower metabolic activities and lack somatic embryogenesis potential. The same results have been found in *Passiflora edulis* Sims. [39] and sugarcane [12]. In addition, the levels of polyphenols and PPO in NEC of Korean pine were significantly higher than those of EC and E-L. Tang et al. [40] found that the accumulation of PPO in Virginia pine can affect the browning of callus and inhibit the growth of callus. This result is expected because phenolics are generally considered to be harmful to in vitro regeneration [13]. It has also been reported that increased PPO activity leads to loss of regenerative capacity and subsequent cell death [30]. Based on this fact, high PPO activity in NEC may indicate a higher level of oxidative stress inside NEC cells, which may inhibit the formation of embryos. However, there was no significant difference in polyphenol content and PPO activity between EC and E-L, which suggested that the loss of somatic embryogenesis potential in E-L may not be related to the accumulation of phenols.

Antioxidant enzymes play an important role in the development of somatic and zygotic embryos, partially due to their ability to remove reactive oxygen species produced by external damage in plants. Moderate stress can cause somatic cells to change their morphology and differentiate [41], but excessive stress can lead to a complete loss of totipotence and eventually cell death [42,43]. In our study, high levels of POD and CAT activity were found in NEC and E-L, while low levels of CAT were found in EC and NEC. Fatima et al. [38] also reported high CAT activity in NEC in a study of somatic embryogenesis in *Catharanthus roseus*. In addition, in a study of wolfberry and gladiolus (*Gladiolus hybrids* Hort.), it was found that CAT activity decreased before SE induction [44]. This indicated that lower levels of hydrogen peroxide metabolism may promote SE differentiation in the early stages of somatic embryogenesis.

### 4.2. Physiological Changes during Somatic Embryo Development of Korean Pine

SE development is a process simulating the development of zygotic embryos [5], in which the maturation of SEs is accompanied by the deposition of storage reserves and a decrease in water content [16]. Our results yielded similar findings to previous studies focused on physiological changes during embryo development, including a study on Maritime pine embryogenesis [24]. The soluble protein content gradually increased during the maturation of Korean pine SEs and reached the highest level in the early cotyledonary stage, indicating that there is a close relationship between soluble proteins and SE differentiation. Several studies have found that storage protein content could be used as a biochemical marker for SE quality since it significantly impacts the success rate of SE germination and survival [45]. It has been reported that somatic embryogenesis depends on carbohydrate metabolism, which can provide energy and carbon for biosynthesis of important molecules during embryo formation [46]. At the same time, carbohydrate is also very important in seed drying and cold tolerance, and acts as a developmental regulator controlling gene expression [24]. High levels of sucrose were also detected during the maturation of Norway spruce SEs [34]. Interestingly, the SE1 line (normal SE development) had higher glucose, fructose and sucrose levels than the SE6 line (inhibited SE development) during Brazilian pine SE maturation. The presence of proteins associated with glycolysis and the tricarboxylic acid cycle in the SE1 cell line reinforces the idea of high energy demand during SE development [47]. This suggested that somatic embryogenesis potential was related to enzymes involved in energy metabolism and starch biosynthesis and degradation.

In many plants, antioxidant enzymes have been shown to play an important role in somatic and zygotic embryogenesis [41]. However, little information is currently available about the role of redox metabolism in somatic embryogenesis. Studies have shown that activity changes in SOD, POD and CAT are related to $O^{2-}$ and $H_2O_2$. $O^{2-}$ that accumulated in cells first reacts with $H^-$ to generate moderately toxic $H_2O_2$ under the catalysis of SOD enzymes. $H_2O_2$ then reacts with H- under the action of enzymes such as CAT to generate $H_2O$ [42]. In our study, the enzyme activities of SOD, POD,

and CAT showed different trends during the maturation of Korean pine SEs. SOD activity increased significantly during the early embryonic development stage (week 1 to week 2), and remained high during the early somatic embryo and cotyledonary somatic stages (week 2 to week 5). Similar results have been found for Prince Rupprecht's larch, where total SOD activity increased by 112% from EC to the globular embryo stage, but did not change further from the globular embryo to cotyledon stage [48]. POD and CAT activities were very high during the first week of culture and then steadily decreased. These trends are similar to those found in a study of somatic embryogenesis in *gladiolus* [49]. In addition, some studies have found that increased SOD activity can promote the differentiation of embryonic cells and early embryo development [50], indicating that changes in SOD activity can be used as an important biochemical indicator during SE maturation. In our study, SOD activity was at a low level during the early somatic embryo period (week 0 to week 1), but increased during the transition from the early somatic embryo period to the cotyledon type embryo period (week 1 to week 5). A similar phenomenon was found during the development of *Catharanthus roseus* SE, where the activity of SOD gradually increased during the development of spheroidal SEs toward heart-shaped SEs [38]. This implies that an increase in SOD activity can promote the differentiation of embryonic cells and the development of early embryos [50]. At the same time, the activities of POD and CAT during the maturation of Korean pine embryos were high in the first week of culture and then began to decline steadily. These trends are similar to those found in Tang somatic embryogenesis studies [49]. In *Catharanthus roseus*, it was also found that the activities of ascorbate peroxidase and CAT decreased rapidly at the initial stage of embryonic development [38]. In addition, a higher level of endogenous reactive oxygen species was observed in the SE1 cell line (with somatic embryogenesis ability) compared to the SE6 cell line (without somatic embryogenesis ability) in Brazilian pine [18,47]. These data indicate that during somatic embryogenesis of Korean pine, SOD, CAT and POD interact to regulate the differentiation and development of SE.

## 5. Conclusions

In summary, our findings provide a new perspective on the factors that affect somatic embryo maturation in cultured Korean pine. High levels of storage materials were found in EC, while high levels of polyphenols and PPO were found in NEC, implying that certain polyphenols may be harmful to somatic embryogenesis. At the same time, high levels of CAT were found in E-L, which may be the reason for E-L's loss of somatic embryogenesis. In addition, during the maturation of Korean pine SE, we found that high storage material content can be used as a physiological indicator of Korean pine SE maturation. The activities of POD and CAT trended upward during early somatic development, while their activity in cotyledon-type embryo remained low. This provides us with a new understanding of the relationship between morphological changes and physiological changes during somatic embryogenesis.

**Author Contributions:** L.Y. and H.S. conceived and designed the study. C.P. and F.G. collected plant materials and prepared SE samples for analysis. C.P. analyzed the results for experiments. L.Y., C.P. and F.G. contributed to the writing of the manuscript and data analyses. H.W. revised the manuscript. All authors read and approved the final manuscript.

**Funding:** The work was supported by the National Key R&D Program of China (2017YFD0600600).

**Acknowledgments:** We thank two anonymous reviewers and the editor for comments that improved an earlier draft of this article.

**Conflicts of Interest:** The authors declare no conflict of interest.

## Abbreviations

| | |
|---|---|
| SE(s) | Somatic embryo(s) |
| EC | Embryogenic callus |
| 2,4-D | 2.4-dichlorophenoxyacetic acid |
| SOD | Superoxide dismutase |
| POD | Peroxidase |
| CAT | Catalase |
| NEC | Non-embryonic |
| E-L | The callus which lost somatic embryogenesis potential |
| PPO | Polyphenol oxidase |
| FW | Fresh weight |
| DW | Dry weight |
| NAA | 1-Naphthaleneacetic acid |
| 6-BA | 6-benzyl amino-purine acid |
| CH | Casein hydrolysate |
| ABA | Abscisic acid |
| NBT | Nitroblue tetrazolium |

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
