# Peer review of "Physiological and Biochemical Traits in Korean Pine Somatic Embryogenesis"

_forests, doi:10.3390/f11050577_

Round 1
Reviewer 1 Report
Please see comments in attached file.

Author Response
Response to Reviewer 1 Comments
Dear Reviewer,
Our sincere thanks to you for the time and effort that you have put into reviewing our manuscript! We found all the comments very constructive and helpful, and have revised our manuscript according to all comments. Please find, below, our point-by-point response to the comments raised.
Thank you for considering our revised manuscript!
Point 1: Substances and enzymatic activities are not morphological markers, but biochemical / physiological markers.
Response 1 Line 20. ‘morphological and physiological’ has been replaced by ‘physiological’.
Point 2: SE involves energy storage, but is not a process of energy storage.
Response 2 Line 25. ‘is a process of’ has been replaced by ‘involves’.
Point 3: In which way the results provide ideas for solving the difficulties of SE in coniferous species? These ideas are not mentioned further in the paper.
Response 3 We edited the sentences as: Lines 27-28, ‘and provide basis for further research on the molecular mechanism of somatic embryogenesis.’
Point 4: “As an important means…” should start a new paragraph.
Response 4 Line 59 ,We have adjusted the paragraph.
Point 5: ““processes” (not “procedures”).
Response 5 Line 62, ‘procedures’ has been replaced by ‘processes’.
Point 6: “Reference needed after “In earlier studies”
Response 6 Line 70, We have added references.
Point 7: Reference needed
Response 7 Line 68,76-77 ,We have added references.
Point 8: The expression “Additionally, the reasons may be…” is not appropriately linked to the sentence before.
Response 8 Line 70-74 , We have modified the connection between sentences.
Point 9:The sentence is unclear: “…maturation seeds-derived callus cells”(?)
Response 9 Line 71-73, We have modified the sentence ‘as the mature seed-derived callus cells contain high levels of lethal phenolics and oxidative compounds that possibly arrest further development of the callus cells’.
Point 10: Line 74. ““Too” (not “to”) high levels…”
Response 10 Line 73 ,‘to’ has been replaced by ‘Too’.
Point 11:Line 80. The reference (12) is wrong (it does not correspond to waht is said in the text).
Response 11 Line 109, We have modified the reference.
Point 12: Line 79-85. This paragraph is not well organized. Soluble protein, sugar, starch, enzymes related to ROS metabolism and morphological changes during SE are all mixed together in an inconsistent discourse.
Response 12 Lines 81-91, We added a section before this paragraph to explain the reason for the physiological index determination.
Point 13: Line 86-95. This paragraph is expected to illustrate the aims and scope of the work, based on what has been told in the Introduction. Nevertheless, morphological differences between different types of calli depending of their SE potential, and morphological changes during maturation of SE are not mentioned in the introduction. The same for fresh and dry weight, and water content.
Response 13 Lines 81-104, We added a description of the purpose and scope of the work in the preface, as follows:
‘ Silveira et al. found that embryogenic callus is smooth and dense in sugarcane, while non-embryonic callus is brittle or soft and translucent [10]. Nieves et al. [11] and Mahmud et al. [12] compared metabolites between sugarcane embryogenic and non-embryonic callus tissues including fats, soluble proteins, amino acids, major carbohydrates, organic acids, phenols and polyamines. According to their results, many of these metabolites have an impact on the somatic embryogenesis process. In a study by Mangosteen, it was found that the embryonic globular structure and the embryogenic nodular compact structure can be used as morphological markers of embryogenesis. A large number of amino acids, as well as mannose, can act as metabolic biomarkers in embryogenic tissues [13]. Therefore, identifying the morphological and physiological differences between embryogenic callus and non-embryonic callus is essential for improving the design of the somatic embryogenesis process.
Somatic embryogenesis is a complex developmental process, which includes cell division, differentiation and a series of physiological changes [14,15]. During somatic embryogenesis of Silybum marianum L., mature somatic embryos were found to contain the highest levels of carbohydrates, starch, ascorbic acid and total free amino acids [16]. In the process of Persian walnut somatic embryogenesis, it was found that CAT, POX, PPO and SOD can not only be used as quantitative and qualitative indicators of the developmental stage of somatic embryos but also reflect the difference between in vivo and in vitro conditions. At the same time, it was also found that translucent cotyledon embryos lack starch granules in epidermal cells compared to opaque cotyledons, while torpedo-shaped somatic embryos contain the highest levels of proline, protein and antioxidant enzyme activity [15]. This demonstrates that understanding the dynamic changes in key biomarkers during the process of somatic embryo differentiation can also reveal the biochemical regulatory mechanisms taking place. Despite the importance of understanding this process, no report has been published on somatic embryogenesis in Korean pine.’
Point 14 Line 89. “This examination provided…” is written as a result, when we should be speaking about aims. In any case, no specific morphological markers are provided in the study.
Response 14 Lines 115-118. The goal of this study is to provide morphological and physiological markers which are correlated with the somatic embryogenesis potential of callus. In addition, morphological and physiological changes which occur in embryogenic cultures during the maturation of Korean pine SEs are examined..
Point15 Line 94-95. The paper reinforces evidence about the relationship between biochemical /physiological markers and SE already described in other species, so, no biochemical markers were “discovered” in the present study.
Response 15 Line 121. ‘discovered’ has been replaced by ‘verified’.
Point 16 Line 99. The reference (14) is wrong. DCR medium was first described by: Gupta, P. K.; Durzan, D. J. Somatic polyembryogenesis from callus of mature sugar pine embryos. Bio/Technology; 1986:4:643– 645.
Response 16 Line 125. We have modified the reference.
Point 17 Line 107. Abbreviation for casein hydrolysate (CH) has not been previously mentioned in the paper.
Response 17 Line 47. We have added the abbreviation of casein hydrolysate.
Point 18 Line 108. “Matte environment” is not clear for me, does it mean not completely in the dark?
Response 18 Line 141. ‘a matte environment’ has been replaced by ‘darkness’.
Point 19 Line 116. This section should be completed with information from the first paragraph of Results section, in “Characterization of analyzed tissues”, and the whole section should be re-written to make it clear in a well organized manner which materials were observed, and times/stages of observation for each of them.
Response 19 Line 150-159. We have described the morphological observation in detail.
‘We used a microscope to observe the morphology of embryogenic callus obtained from immature seeds, non-embryonic callus obtained from mature zygote embryos and callus which had lost somatic embryogenesis potential. EC was collected on the 7th day of proliferation culture, stained with 2% acetylcarmine, compressed on slides, covered with cover glass, and observed under an Zeiss A× microscope (Carl Zeiss, Jena, Germany). In addition, we also observed the morphological changes during somatic embryogenesis of Korean pine under a microscope (SZX-ILLB2-200, Olympus Corporation, Tokyo, Japan). The developmental period of SEs of Korean pine was defined according to the results of Von Arnold et al. [23]. Materials were collected for morphological observation at 1, 2, 3, 4, 5 and 8 weeks of somatic embryo maturation and 2 months of somatic embryo germination.’
Point 20 Line 117-118. As samples from the three types of calli are taken, it should say: “the calli/the samples were stained…”
Response 20 Line 150-159. This part has been solved in the last question.
Point 21 Line 121-126. As far as I understand, samples from EC, after two weeks in proliferation medium, were transfered to maturation medium, and samples were taken at 0, 1, 2, 3, 4, 5 weeks. The explanation in the text should be clearer.
Response 21 Line 161-166 .We have described the steps of material collection in detail.
‘For EC, NEC and E-L, three biological replicates were collected on the 7th day of subculture, and 0.5 g of callus was collected for each sample for physiological examination. ECs were collected for somatic embryo maturation on the 7th day of subculture. During somatic embryo maturation, three biological replicates were collected at 0, 1, 2, 3, 4 and 5 weeks, with 0.5 g obtained for each sample for physiological analysis. These samples were then wrapped in aluminum foil, immediately frozen in liquid nitrogen and stored at -80 °C until further analysis.’
Point 22 Lin 133. Reference for anthrone method should be included: Somani BL, Khanade J, Sinha R A modified anthrone-sulfuric acid method for the determination of fructose in the presence of certain proteins. Anal Biochem. 1987 Dec; 167(2):327-30.
Response 22 Line 176. We have added literature.
Point 23 Line 143. Colvin et al. [rerefence number is lacking] (year must be removed)
Response 23 Line 190. We have added literature.
Point 24 Line 149. Liu et al.
Response 24 Line 196. ‘Liu’ has been replaced by ‘Liu et al.’
Point 25 Line 156. Is it 5 mL or 0.5 mL?
Response 25 Line 203. Here is 5 mL.
Point 26 Line 158. Better, “Folin-Cioalteu (phenol) reagent”
Response 26 Line 205. ‘Folin-Ciocalteu’ has been replaced by ‘Folin-Cioalteu (phenol) reagent’.
Point 27 Line 161. Helaly et al.
Response 27 Line 209. ‘Helaly’ has been replaced by ‘Helaly et al’.
Point 28 Line 171. Nitroblue tetrazolium abbreviation (NBT) should be included, as it will be used in line 173.
Response 28 Line 49 .We have abbreviated nitroblue tetrazolium abbreviation (NBT).
Point 29 Line 172. Jariteh et al.
Response 29 Line 220. ‘Jariteh’ has been replaced by ‘Jariteh et al.’
Point 30 Line 175. The reference for Abeles and Biles (1991) is wrong. Although several papers cite this work for the method described, it does not correspond to the one described by Abeles and Biles in their article. I suggest to use the following reference for the method: H. Rahnama, H. Ebrahimzadeh The effect of NaCl on antioxidant enzyme activities in potato seedlings. Biologia Plantarum, 49 (2005), pp. 93-97.
Response 30 Line 224. We have revised the literature.
Point 31 Line 180. Jariteh et al.
Response 31 Line 228. ‘Jariteh’ has been replaced by ‘Jariteh et al.’
Point 32 Line 184. Kaewubon et al.
Response 32 Line 232. ‘Kaewubon’ has been replaced by ‘Kaewubon et al.’
Point 33 Line 192. Which statistical analyses were performed using Microsoft Excel 2007?
Response 33 Line 239 .We have modified the data analysis part.
Point 34 Line 197-211: All this section is confusing and should be re-written. From line 198 to 202 (until “…(Figure 1 b)”), there is a description which should be included in Materials and Methods. After that, the rest of the paragraph is not well organized and should be re-written (including correction of english language). The last sentence is clearly wrong: “…no somatic embryo development Seedling regeneration” (?).
Response 34 Line 244-251. We incorporate the "From line 198 to 202 (until" ... (Figure 1 b) ")," part into materials and methods. And rewrite the rest.
Point 35 Line 224-227. Attention! According to Figure 2, this fragment is plenty of wrong interpretations: Contrarily to what is stated: 1) polyphenol content, and PPO and SOD activities of EC are lower than that of NEC but not than that of E-L; 2) there are no significant differences in CAT activity between EC and NEC. E-L and EC starch content does not differ significantly, E-L PPO activity is not significantly different to EC (!)
Response 35 Line 263-270. Based on the analysis of the results of this part, we modify the content as follows ‘There were significant differences in soluble protein, soluble sugar, starch, proline, polyphenols, PPO, POD, SOD and CAT among Korean pine EC, NEC and E-L (PË‚0.05). Compared with NEC, EC had higher soluble protein, soluble sugar, starch and proline content, and lower polyphenol, POD, SOD and POD activities. Compared with E-L, EC had higher soluble protein, soluble sugar and proline content, and lower POD and CAT activity. However, there was no significant difference between starch content, polyphenol content, PPO and SOD activity. Compared with E-L, NEC had higher polyphenol content, PPD and SOD activity, lower starch content and CAT activity, but no significant difference in soluble protein, soluble sugar, proline content or POD activity. (Figure 2).’
Point 36 Line 228. Figure 2 (H): letters expressing significant differences are lacking.
Response 36 Line 271-272. We have added the missing letters in Figure 2.H.
Point 37 Line 234-237. The first part of the paragraph, until “…to maturation medium” corresponds to Materials and Methods.
Response 37 Lines 129-138. The first part of this paragraph has been deleted and added in the method.
Point 38 Line 242. Figure 3. Standardizing the bar lenght in a fixed value would permit to compare relative sizes along the process.
Response 38 Lines 284. We modified the bar lenghtin Figure 3.
Point 39 Line 248. Measurements of FW, DW and water content are not described in Materials and Methods. On the other hand, as the meaning of FW and DW is not appropriately defined (Figure 4), an apparent contradiction arises: if FW is x 6.8 and DW is x 3.7 in week 5 as compared to week 0, how can water content be lower?? The explanation to this lies in the fact that, for the authors, DW is expressed in terms of % of FW, and not in grams.
Response 39 Lines 295, 168. We changed the unit of DW to g, and changed to draw Figure 4 first. The measurement methods and calculation methods of FW, DW, and Water content were added to the materials and methods.
Point 40 Line 257-258. The statement that the highest levels of soluble sugar are observed at week 5 is wrong, Considering the results from Figure 4, statistically, there is no difference in sugar content from week 2 to week 5.
Response 40 Lines 300-305. We have modified the analysis of soluble sugar content as follows.
‘The highest levels of soluble sugar (10.4 mg g-1 FW), soluble protein (10.72 mg g-1 FW) and starch (12.52 mg g-1 FW) were detected in the fifth week of embryogenesis, and the lowest levels were detected at week 0. Relative to week 0, the developing embryos in the fifth week had 51.12%, 43.90% and 73.89% higher levels of soluble sugar, soluble protein and starch, respectively (Figure 5).’ modified to ‘The highest level of soluble sugar (10.40 mg g-1 FW) was detected in the second week of embryogenesis, which was 1.78 times higher than that in week 0. In addition, there was no significant change in soluble sugar content from week 2 to week 5. The highest levels of soluble protein (10.72 mg g-1 FW) and starch (12.52 mg g-1 FW) were detected in the 5th week of embryogenesis. Compared to week 0, the soluble protein and starch content increased by 3.95 and 3.82 fold (Figure 5).’
Point 41 Line 266-268. In contradiction with what is stated, the level of SOD activity does not vary significantly from week 2 to week 5.
Response 41 Lines 310-314. We have modified the analysis of SOD activity as follows.
‘SOD activity significantly increased during the maturation of SEs and the highest level of SOD activity (0.33 U min-1 mg-1 protein) was detected in the fifth week, which was 94.11% higher than that detected at week 0” modified to “During the maturation of SEs, SOD activity increased significantly at the early stage of somatic embryo development (week 1 to week 2). At its peak, SOD activity was as high as 0.33 U min-1 mg-1 protein, an increase of 78.41% over week 0. The SOD activity was consistently high from early somatic embryos to cotyledonary embryos (week 2 to week 5)’
Point 42 Line 268-272. Contrarily to what is stated, there is no significant increase in the first week for POD and CAT activities, and CAT activity is not significantly different in week 4 and 5.
Response 42 Lines 314-319. We have modified the analysis of POD activity and CAT activity as follows.
‘The activities of POD and CAT increased during the first week of SE maturation, followed by a downward trend. The lowest level of POD activity (0.34 U min-1 mg-1 protein) was detected in the fourth week, which was 68.22% lower than the level at week 0. The lowest level of CAT activity (0.51 U min-1 mg-1 protein) was detected in the fifth week, which was 46.32% lower than the activity level at week 0 (Figure 5). “ modified to “POD activity was high during the early somatic stage (week 0 to week 2), and gradually decreased during the transition from early somatic stage to cotyledonary embryo (week 2 to week 5). POD activity at week 4 was reduced by 68.22% compared to week 2 (0.34U min-1 mg-1 protein). CAT activity was high during early somatic development (week 0 to week 3), began to decline at week 3, and dropped to 0.54 U min-1 mg-1 protein by week 4 (Figure 5).’
Point 43 Line 279-280. The sentence “…lost SEs production after one year of subculture ability” must be revised. Written in this manner: “…lost SEs production ability after one year of subculture” it would have a sense.
Response 43 Lines 326. ‘lost SEs production after one year of subculture ability” has been replaced by “lost SEs production ability after one year of subculture’.
Point 44 Line 280. Reference [26] is not about Pinus nigra, but about peach palm (!).
Response 44 Lines 327. ‘Pinus Nigra' has been replaced by ‘Pinus nigra and peach palm’.
Point 45 Line 283. In the text fragment “…we compared three types of physiological differences.”, what are you refering to?
Response 45 Lines 330. We compared the physiological differences between EC, NEC and E-L.
Point 46 Line 294. It should be taken into consideration that there is a vast range of phenolic compounds, and some of them are cofactors of in vitro organogenic phenomena. On the other hand, the increase in polyphenol levels can be (in part) a consequence of the lack of embryogenetic response and not necessarily the cause.
Response 46 Lines 340-342. We have modified the influence of phenols.
Point 47 Line 304. CAT activity is not significantly different in EC and NEC (already commented).
Response 47 Lines 353. ‘EC’ has been replaced by ‘EC and NEC’.
Point 48 Line 305. Fatima et al. [29]
Response 48 Lines 353. ‘Fatima’ has been replaced by ‘Fatima et al. [29]’.
Point 49 Line 318. “Biochemical marker”, better than “molecular marker”.
Response 49 Lines 366. ‘molecular marker’ has been replaced by ‘Biochemical marker’.
Point 50 Line 337. The increase of SOD activity is observed just between week 1 and 2, so it should not be described as a “gradual” increase along the process.
Response 50 Lines 384-386. ‘The SOD activity gradually increased during the maturation of Korean pine SEs and stayed at a high level throughout the process.’ has been replaced by ‘During the maturation of Korean pine SEs, “SOD activity increased significantly during the early embryonic development stage (week 1 to week 2), and remained high during the early somatic embryo and cotyledonary somatic stages (week 2 to week 5).”
Point 51 Line 347. The reference cited (29) is about Catharanthus roseus, and not about Vinca (!)
Response 51 Lines 357. ‘Vinca’ has been replaced by ‘Catharanthus roseus’.
Point 52 Line 354-55. It should not be concluded from the study that the physiological differences found in the different types of calli are the cause of the different responses in terms of somatic embryogenesis potential/ability. On the other hand, the relationship between morphological and physiological changes was not treated in the study. To my opinion, conclusions should focus on the fact that the observations made in Korean pine in this study are new in this species, and are in line with observations made for other species, which reinforces the existing hypotheses for the role of the different physiological markers studied in somatic embryogenesis. Further considerations could be considered as speculative.
Response 52 Lines 411-420 .We have revised the conclusions as follows.
In summary, our findings provide a new perspective on the factors that affect somatic embryo maturation in cultured Korean pine. High levels of storage materials were found in EC, while high levels of polyphenols and PPO were found in NEC, implying that certain polyphenols may be harmful to somatic embryogenesis. At the same time, high levels of CAT were found in E-L, which may be the reason for E-L's loss of somatic embryogenesis. In addition, during the maturation of Korean pine SE, we found that high storage material content can be used as a physiological indicator of Korean pine SE maturation. The activities of POD and CAT trended upward during early somatic development, while their activity in cotyledon-type embryo remained low. This provides us with a new understanding of the relationship between morphological changes and physiological changes during somatic embryogenesis.
Point 53 English language must be improved. Sometimes, defficient english make the text confusing for the reader.
Response 53 We have accepted your comments and we revised the English in the article. Since we are not native English speakers, the English text of a draft of this manuscript was proofread by TopEdit (www.topeditsci.com). We have two certificates.
Thank you again!
Sincerely yours,
Ling YANG and Hai-long SHEN

Reviewer 2 Report
I have attached a file (Forests review 4-20) for my comments and suggestions.

Author Response
Response to Reviewer 2 Comments
Dear Reviewer,
Our sincere thanks to you for the time and effort that you have put into reviewing our manuscript! We found all the comments very constructive and helpful, and have revised our manuscript according to all comments. Please find, below, our point-by-point response to the comments raised.
Thank you for considering our revised manuscript!
Point 1: Line102. Should “zygotic” be replaced by “somatic”? The embryogenic cell line is called as ECM28-10 and the non-embryogenic tissue is NECM28-10, which indicates these two types of tissue were from the same cell line, or the same genotype. If non-embryogenic tissue was induced from mature zygotic embryos, these two types of tissue could not be the same genotype and the author need to explain how to avoid genotype effects by experimental design and data analysis.
Response 1 Lines 130. We have considered whether "somatic cells" can replace "zygote embryos", but comprehensive experimental materials, we feel that "zygote embryos" are more suitable for this article. 2. In our study, mature seeds and immature seeds come from the same open pollinated mother tree. We have added in materials and methods.
Point 2: Lins 252, 254. Both “moisture content” and water content” were used. It is better to keep consistency. In addition, what is the formula used for water content (with reference)? In my knowledge, water content (%) = (FW-DW) / FWx100
Response 2 Line 294. ‘moisture content’ has been replaced by ‘water content’ and the calculation method of water content has been added to 2.5 of the materials and methods.
Point 3: L. 74 to high - too high; 2.4 – 2,4
Response 3 Line 73. ‘to high’ has been replaced by ‘too high’ and ‘2.4’ has been replaced by ‘2,4’.
Point 4: L. 98 DCR - need full name at the first place 4
Response 4 Line 125. We considered whether DCR was used comprehensively, but at the same time, considering the widespread use of DCR medium in tissue culture, we did not modify it.
Point 5: L. 108 matte environment?
Response 5 Line 141. ‘matte environment’ has been replaced by ‘darkness’.
Point 6: L. 113 0.2 M sucrose - ? g/ L - need consistency
Response 6 Line 147. ‘0.2 M sucrose’ has been replaced by ‘68 g L-1 sucrose’.
Point 7: L. 120 was taken. - was taken with what?
Response 7 Lines 148-157. In this section, we have a new description of morphological observations. Modify as follows.
‘We used a microscope to observe the morphology of embryogenic callus obtained from immature seeds, non-embryonic callus obtained from mature zygote embryos and callus which had lost somatic embryogenesis potential. EC was collected on the 7th day of proliferation culture, stained with 2% acetylcarmine, compressed on slides, covered with cover glass, and observed under an Zeiss A× microscope (Carl Zeiss, Jena, Germany). In addition, we also observed the morphological changes during somatic embryogenesis of Korean pine under a microscope (SZX-ILLB2-200, Olympus Corporation, Tokyo, Japan). The developmental period of SEs of Korean pine was defined according to the results of Von Arnold et al. [23]. Materials were collected for morphological observation at 1, 2, 3, 4, 5 and 8 weeks of somatic embryo maturation and 2 months of somatic embryo germination.’
Point 8: L. 132 took - grammar
Response 8 Line 179. ‘took’ has been replaced by ‘was taken’.
Point 9: L. 280 delete “ability”
Response 9 Lines 326. ‘while EC obtained from immature seeds lost SEs production after one year of subculture ability’ has been replaced by ‘while EC obtained from immature seeds lost SEs production ability after one year of subculture’.
Thank you again!
Sincerely yours,
Ling YANG and Hai-long SHEN

Reviewer 3 Report
The paper presents valuable data and brilliant pictures. I think a careful and extended revision is required to increase the scientific value.
- The aim is rather general formulated and this hinders a detailed discussion of the results. Moreover, in the title the focus is on physiological characterization of callus. For this, less data are presented (only one date of analyses after 7 days) than for the embryo development/maturation (6 dates of analyses from week 1 to week 6). The title should focus the main outcome of the research and therefore title should be changed or more data for callus stage should be added.
- The information regarding plant material is to complete and to make easier understandable: Are the seed for all variants from the same tree (28-10)? Otherwise, the genetic effect should be also discussed not only the physiological ones. The immature and mature embryos used as starting material should be described to make experiment repeatable.
- The experimental outline should be more clearly explained: Which medium, how long on this medium needs to be added.
- It should be clear how many replicates there are per variant.
- You mention a physiological index (2.4.) but it is not described what you mean or how you calculated. Later on you did not use this index. I guess this could be interesting to work on.
- In figures, all information regarding medium and week should be added and the order of graphs in Figure 2 and 5 should be the same.
- In the discussion, you should focus more on your results and avoid mix up with literature and too much speculation.
- For callus, you analyzed a status after 7 days this is not an activity! An analysis for several days would be necessary to learn something about the activity.
- You use a lot of literature but especially in the discussion regarding stress you refer often on quite different plants. Maybe you can find literature at least for conifers.
- line 17- 19 “These results indicate that high contents of storage substance (protein, sugar and starch), and low levels of polyphenols and polyphenol oxidase can be used as morphological and physiological markers of callus with somatic embryogenic potential.”
Remark to that: These analyses were done only ones in callus cultures. There is no time course of behavior of these compounds shown. Such data are presented for maturation phase of the SEs. These are valuable data but not sufficient for discrimination of embryogenic and non-embryogenic lines. Moreover, the proof that non-embryogenic lines do not form embryos is missing, because this callus was not transferred to the next medium. It should be also explained why just after 7 days these analyses were conducted. Is there literature regarding this?
- Line 336: “The SOD activity gradually increased during the maturation of Korean pine SEs and stayed at a high level throughout the process.”
Remark: yes, but only up to two weeks, please be precise and try to link the enzyme activity with embryo developmental stages. There are different patterns for the three enzymes!
- Chapter 3.5 . You measured content of protein, sugar and starch based on fresh weight, however, in embryo development in the same time the water content is reduced, please check, whether the content in dry matter is really higher. Data based on fresh matter indicate probably a higher increase than it is.
Some more remarks you find on the manuscript.

Author Response
Response to Reviewer 3 Comments
Dear Reviewer,
Our sincere thanks to you for the time and effort that you have put into reviewing our manuscript! We found all the comments very constructive and helpful, and have revised our manuscript according to all comments. Please find, below, our point-by-point response to the comments raised.
Thank you for considering our revised manuscript!
Point 1: Line 74. “Too” (not “to”) high levels…”
Response 1: Line 73. ‘to’ has been replaced by ‘Too’.
Point 2: The aim is rather general formulated and this hinders a detailed discussion of the results. Moreover, in the title the focus is on physiological characterization of callus. For this, less data are presented (only one date of analyses after 7 days) than for the embryo development/maturation (6 dates of analyses from week 1 to week 6). The title should focus the main outcome of the research and therefore title should be changed or more data for callus stage should be added.
Response2: Line 2. In response to this problem, we have revised the title of the article to ‘
The impact of physiological and biochemical properties on somatic embryogenesis in Korean pine’.
Point 3: The information regarding plant material is to complete and to make easier understandable: Are the seed for all variants from the same tree (28-10)? Otherwise, the genetic effect should be also discussed not only the physiological ones. The immature and mature embryos used as starting material should be described to make experiment repeatable.
Response 3: Lines 125-133. In response to this problem, we provided a detailed introduction of immature seeds and mature seeds in the materials and methods section. In this study, immature seeds and mature seeds are from the same open pollinated mother tree.
Point 4: The experimental outline should be more clearly explained: Which medium, how long on this medium needs to be added.
Response 4: Lines 125-142. We added specific materials and cultivation time in the materials and methods section.
Point 5:It should be clear how many replicates there are per variant.
Response 5: Lines 161-166. We supplemented the number of repetitions of each measurement index in the materials and methods.
Point 6: You mention a physiological index (2.4.) but it is not described what you mean or how you calculated. Later on you did not use this index. I guess this could be interesting to work on.
Response 6: Line 162. Here we have replaced ‘Physiological index determination’ with ‘Physiological determination’, and carried out a detailed description of these physiological determinations in the ‘2.6.Physiological determination’ section.
Point 7: In figures, all information regarding medium and week should be added and the order of graphs in Figure 2 and 5 should be the same.
Response 7: Lines 272, 307. We modified the captions of Figure 2 and Figure 5, added the medium matrix and culture time, and also modified the serial number in Figure 5. Modify as follows.
Figure 2. Physiological differences between different callus tissues of Korean pine on the 7th day of culture on proliferation medium.
Figure 5. Physiological changes of somatic embryos of Korean pine in maturation medium at 0, 1, 2, 3, 4 and 5 weeks
Point 8: In the discussion, you should focus more on your results and avoid mix up with literature and too much speculation.
Response 8 : Lines 384-408. We have revised the discussion section.
Point 9: For callus, you analyzed a status after 7 days this is not an activity! An analysis for several days would be necessary to learn something about the activity.
Response 9: Lines 143. We are very interested in the physiological changes of the tissue during the subculture, and we will consider related research in future research.. However, in this study, we chose embryogenic callus on the 7th day of subculture to induce somatic embryo maturation, so for the physiological analysis of the three callus, we chose the callus on the 7th day of subculture Determination.
Point 10: You use a lot of literature but especially in the discussion regarding stress you refer often on quite different plants. Maybe you can find literature at least for conifers.
Response 10: We added research related to pine trees in the discussion section. The added documents are [41], [34], [49].
Point 11: line 17- 19 “These results indicate that high contents of storage substance (protein, sugar and starch), and low levels of polyphenols and polyphenol oxidase can be used as morphological and physiological markers of callus with somatic embryogenic potential.”
Response 11: We did not conduct experiments on non-embryogenic callus tissues unable to form somatic embryogenesis, because non-embryonic tissues gradually become hard and yellow during the proliferation process, and their proliferation ability is weakened. In future research, we will consider the reasons why non-embryonic callus cannot form somatic embryos.
Point 12: Line 336: “The SOD activity gradually increased during the maturation of Korean pine SEs and stayed at a high level throughout the process.”
Response 12 : Lines 384-408. In the discussion section, we first modified the discussion on SOD and CAT, and combined the changes in enzyme activity with the morphological changes to elaborate.
Point 13: Chapter 3.5. You measured content of protein, sugar and starch based on fresh weight, however, in embryo development in the same time the water content is reduced, please check, whether the content in dry matter is really higher. Data based on fresh matter indicate probably a higher increase than it is.
Response 13: Lines 168-173, 300-305. We have modified the unit of DW, and now the results show that DW gradually increases with the maturation of somatic embryos.
Point 14: Unfortunately this data are not available because not yet published. However, they are important for the presented work. Please, find a solution.
Response 14 : Lines 124-142. We added the method of somatic embryo maturation in Korean pine.
Piont 15: Literature? Otherwise it is a postulation.
Response15: Lines 76-77. We have added literature
Point 16: in Pinus or in general?
Response 16: general
Point 17: By whom it was postulated?
Response 17: The phrase ‘It has been postulated that the state of callus may determine the ability of ECs to undergo somatic embryogenesis’ has been deleted
Point 18: Is there no literture regarding this for conifers?
Response 18: Lines109. We have added literature on conifers.[18]
Point 19: this is a conclusion, however, it is too early for this conclusion because the callus is only ones investigated.
Response 19: Lines 118-122 .Our modifications are as follows.
The biomarkers verified in this study lay a foundation for understanding the molecular mechanisms of somatic embryogenesis in Korean pine.
Point 20: type, company?
Response 20: Line 154. We have added
Point 21: but you had also early SE stages on this picture.
Response 21: Line 252. The callus obtained in mature seeds has a morphology similar to EC at the beginning but changes with the proliferation of culture. We will study this phenomenon in future research.
Thank you again!
Sincerely yours,
Ling YANG and Hai-long SHEN
Round 2
Reviewer 1 Report
The manuscript has been drastically improved. For minor specifical comments and suggestions please see attached file.

Author Response
Response to Reviewer 1 Comments
Dear Reviewer,
Our sincere thanks to you for the time and effort that you have put into reviewing our manuscript! We found all the comments very constructive and helpful, and have revised our manuscript according to all comments. Please find, below, our point-by-point response to the comments raised.
Thank you for considering our revised manuscript!
Point 1: The "impact"... was not specifically studied. I would propose: "Physiological and biochemical traits in Korean pine somatic embryogenesis".
Response 1: Line 2. Thanks to the reviewers for their comments, we decided to use "Physiological and biochemical traits in Korean pine somatic embryogenesis" as the topic
Point 2: non-embryogenic
Response 2: Line 15. ‘Non-embryonic callus’ has been replaced by ‘non-embryonic callus’
Point 3: substances
Response 3: Line 15, 18. ‘substance’ has been replaced by ‘substances’
Point 4: and
Response 4: Line 24 .‘where’ has been replaced by ‘and’
Point 5:mechanisms
Response 5: Line 27. ‘mechanism’ has been replaced by ‘mechanisms’
Point 6: Non-embryogenic
Response 6: Line 39. ‘Non-embryonic callus’ has been replaced by ‘Non-embryogenic’
Point 7: which
Response 7: Line 40. ‘of’ has been replaced by ‘which’
Point 8: Nitroblue tetrazoliumabbreviation
Response 8: Line 48. ‘Nitroblue tetrazoliumabbreviation’ has been replaced by ‘Nitroblue tetrazoliu’
Point 9: but
Response 9: Line 61. ‘and’ has been replaced by ‘but’
Point 10: mechanisms
Response 10: Line 64. ‘mechanism’ has been replaced by ‘mechanisms’
Point 11: acid
Response 11: Line 124. ‘acids’ has been replaced by ‘acid’
Point 12:a
Response 12: Line 150. ‘an’ has been replaced by ‘a’
Point 13: determinations
Response 13: Line 170. ‘determination’ has been replaced by ‘determinations’
Point 14: revise this fragment
Response 14: Lines 180-185. We have revised this paragraph as follows.
Extraction of starch was modified according to the method of Vale et al. [26]. The particles from soluble carbohydrate extract were resuspended in 1 mL of perchloric acid (9.2 mol·L-1), stirred for 15 minutes, and then Centrifuge at 12,000 g for 10 minutes at 4° C and collect the supernatant. The resulting precipitate was then resuspended in 1 mL of perchloric acid (4.6 mol·L-1), stirred for 15 minutes, and centrifuged at 12,000 g for 10 minutes at 4 ° C, and the supernatant was collected. The two collected supernatants were mixed and brought up to 50 mL.
Point 15: were
Response 15: Line 187. ‘was’ has been replaced by ‘were’
Point 16: multiplying
Response 16: Line 189. ‘multiplyed ’ has been replaced by ‘multiplying’
Point 17: NEC
Response 17: Line 242. ‘NEC’ has been replaced by ‘NECs’
Point 18: embryogenic
Response 18: Line 255. ‘Non-embryonic’ has been replaced by ‘Non- embryogenic’
Point 19: In addition
Response 19: Line 296. We have deleted ‘ In addition’
Point 20: Catharanthus roseus
Response 20: Lines 390, 396. ‘Catharanthus roseu’ has been replaced by ‘Catharanthus roseus’

Reviewer 3 Report
The authors improved the scientific quality by revision considerably. There are few yellow labelled words shoulb be checked again before publishing.

Author Response
Response to Reviewer 3 Comments
Dear Reviewer,
Our sincere thanks to you for the time and effort that you have put into reviewing our manuscript! We found all the comments very constructive and helpful, and have revised our manuscript according to all comments. Please find, below, our point-by-point response to the comments raised.
Thank you for considering our revised manuscript!
Point 1: not only
Response 1: line 60. ‘This process not only provides’ has been replaced by ‘This not only process provides’.
Point 2: You used these terms already in line 95 . Please give the full term in case of the first use of abbreviation.
Response 2: lines 94- 95. We have made amendments to use the full term for the first time.
